# Cytolysin A (ClyA): A Bacterial Virulence Factor with Potential Applications in Nanopore Technology, Vaccine Development, and Tumor Therapy

**DOI:** 10.3390/toxins14020078

**Published:** 2022-01-21

**Authors:** Kazunori Murase

**Affiliations:** Department of Microbiology, Graduate School of Medicine, Kyoto University, Kyoto 606-8501, Japan; murase.kazunori.3x@kyoto-u.ac.jp; Tel.: +81-75-753-4448

**Keywords:** cytolysin A, pore-forming toxin, outer membrane vesicles, biotechnology application

## Abstract

Cytolysin A (ClyA) is a pore-forming toxin that is produced by some bacteria from the *Enterobacteriaceae* family. This review provides an overview of the current state of knowledge regarding ClyA, including the prevalence of the encoding gene and its transcriptional regulation, the secretion pathway used by the protein, and the mechanism of protein assembly, and highlights potential applications of ClyA in biotechnology. ClyA expression is regulated at the transcriptional level, primarily in response to environmental stressors, and ClyA can exist stably both as a soluble monomer and as an oligomeric membrane complex. At high concentrations, ClyA induces cytolysis, whereas at low concentrations ClyA can affect intracellular signaling. ClyA is secreted in outer membrane vesicles (OMVs), which has important implications for biotechnology applications. For example, the native pore-forming ability of ClyA suggests that it could be used as a component of nanopore-based technologies, such as sequencing platforms. ClyA has also been exploited in vaccine development owing to its ability to present antigens on the OMV surface and provoke a robust immune response. In addition, ClyA alone or OMVs carrying ClyA fusion proteins have been investigated for their potential use as anti-tumor agents.

## 1. Introduction

Toxins play an important role in pathogenicity in the host–pathogen interaction, and especially during the process of infection. These virulence determinants can inhibit protein synthesis, help the pathogen evade host immune responses, and cause damage to the target cell membrane in multiple ways [1,2]. Depending on the nature, structure, and mode of action, bacterial toxins can be classified into several functional groups [3,4]. The largest group of protein toxins is the pore-forming toxins (PFTs), which form pores on a target cell membrane, increasing membrane permeability and creating an ion imbalance [5,6,7]. Based on the secondary structures of the domains that form the transmembrane pores, PFTs can be broadly classified into two structural families [8,9]: those containing helices (α-PFT), such as the colicins produced by *Escherichia coli* [10], and those containing sheets (β-PFT), such as alpha-hemolysin produced by *Staphylococcus aureus* [11].

Cytolysin A (ClyA, also known as HlyE or SheA) is a 34-kDa α-PFT that is produced by some bacteria from the *Enterobacteriaceae* family, including *E. coli* and *Salmonella enterica*. This toxin was first reported at around the same time by both the Ludwig and Oscarsson groups, who observed that overexpression of the *Salmonella enterica* subsp. *enterica* serovar Typhimurium regulatory protein SlyA in the *E. coli* strain K-12 resulted in the synthesis of a novel hemolysin/cytolysin that conferred a hemolytic phenotype on the recombinant *E. coli* strain [12,13]. Since its initial discovery, the structure and function of the ClyA toxin, as well as its role as a virulence factor, have been explored in numerous studies. Structural studies using X-ray crystallography and electron microscopy have shown that ClyA is one of only a few α-PFTs that can exist as both a soluble monomer and as a transmembrane pore (undergoing large-scale conformational changes) [14,15,16]. These findings provided detailed structural information that has accelerated the subsequent structural analysis of other α-PFTs [17,18,19]. Another remarkable finding is that ClyA is secreted from bacterial cells via a vesicle-mediated pathway [20]. Currently, one of the most intriguing research areas regarding this toxin is its potential utility in applied research, such as in the clinical setting. A recent report demonstrating that recombinant ClyA can be displayed on the surface of bacterial cells or released by membrane vesicles suggests that ClyA could be used in applications such as vaccine development and tumor therapy.

This review first presents a brief summary of the current state of knowledge regarding ClyA (including the prevalence of the encoding gene and its transcriptional regulation, the secretion pathway used by the protein, and the mechanism of protein assembly into the final form). The unique functional features of ClyA are highlighted, and recent research exploring its potential use in biotechnology applications (vaccine development, tumor therapy, and nanopore formation) are discussed.

## 2. General Features of ClyA

### 2.1. Prevalence of the clyA Gene in Bacteria

The *clyA* gene is a chromosomal gene that was first identified in the *E. coli* strain K-12 [12,13]. Although the gene is widely distributed in various *E. coli* strains, sequencing analysis showed that it was disrupted in half of the tested strains, and that the defective forms could be classified into several inactivation patterns based on different insertions and/or deletions [21,22,23]. In addition, this inactivated *clyA* gene is found exclusively in *E. coli* phylogroup B2 strains, including enteropathogenic *E. coli* and extraintestinal pathogenic *E. coli*, while the intact gene is relatively conserved in strains from other phylogroups [23].

Homologs of ClyA are also found in the pathogenic bacteria *Salmonella* and *Shigella* [24,25]. *Shigella* strains often harbor nonfunctional copies of *clyA* that have been inactivated either by the integration of insertion sequence elements (*S. dysenteriae*, *S. boydii*, and *S. sonnei*) or by a frameshift mutation (*S. flexneri*), similar to the inactivated forms of *clyA* observed in *E. coli* [26]. Regarding *Salmonella*, von Rhein et al. reported that the *S.* Typhi and *S.* Paratyphi A strains they tested each harbored an intact copy of *clyA* [27]. However, it has been shown that the *clyA* gene is absent in strains of *S. enterica* serovar Paratyphi B and serovar Paratyphi C, in various non-typhoidal *S. enterica* subsp. *enterica* serovars (Typhimurium, Enteritidis, Choleraesuis, Dublin, and Gallinarum), and in *S. enterica* subsp. *arizonae* and *Salmonella bongori*. Thus, the *clyA* gene and its homologs are conserved in some bacteria from the *Enterobacteriaceae* family.

### 2.2. Transcriptional Regulation

The transcriptional regulation of *clyA*, which has been well-studied, is somewhat complex in that it involves several known regulators and is subject to the influence of environmental factors (Table 1).

Transcription of *clyA* is normally prevented by the heat-stable nucleoid-structuring (H-NS) protein [28,29], which is a global transcriptional regulator that influences the expression of approximately 5% of all genes in *E. coli*. The direct influence of H-NS on transcription is almost universally negative, often regulating gene expression in response to environmental conditions [30,31,32]. Therefore, *clyA* transcription is only derepressed and activated in H-NS-deficient strains, which is why ClyA is sometimes called the “silent hemolysin”. In addition to derepression in the context of H-NS deficiency, *S.* Typhimurium SlyA and *E. coli* MprA, both of which are members of the MarR family of transcriptional regulators, activate *clyA* transcription [12,13,24]. H-NS and SlyA share a binding site in the *clyA* promoter region; therefore, overexpression of SlyA can activate *clyA* expression by antagonizing H-NS-mediated transcriptional repression [28,33]. It has recently been reported that (p) ppGpp, an important nucleotide that acts as a secondary messenger in the stringent response to stress, also affects *clyA* expression in a SlyA-independent manner [34]. Two other transcriptional regulators, cyclic AMP receptor protein (CRP) and fumarate and nitrate reduction regulator (FNR), also positively regulate *clyA* expression by binding at the same site in response to two distinct environmental signals, oxygen starvation (for FNR) and glucose starvation (for CRP) [23,29,33,35]. The ClyA-derived hemolytic activity of *E. coli* has been shown to be strongly enhanced under anaerobic conditions [23]. In *S.* Typhi, *clyA* is part of an operon that also contains the *taiA* gene and is located in the *Salmonella* pathogenicity island, and *clyA* expression is activated by the PhoPQ two-component system [36]. Another study showed RpoS-dependent transcriptional upregulation at a low pH or high osmolarity, as well as oxygen depletion-independent regulation, of *clyA* [37]. In addition, CRP can downregulate *clyA* expression in *S.* Typhi [37]. Later work by Jofre indicated that RpoS is the central regulator in the *clyA* regulatory network, integrating the CRP and PhoP signaling pathways [38]. Thus, the mechanisms regulating *clyA* expression seem to differ among bacterial species, at least between *E. coli* and *S.* Typhi. In addition, *clyA* expression is modulated by several global regulators via a variety of complicated mechanisms and is influenced by multiple environmental signals.

### 2.3. Secretion Pathway

Bacterial toxin transport across bacterial membranes to reach their targets is mediated by a variety of cotranslational and posttranslational modifications. Toxin transport occurs by multiple mechanisms, which have been characterized in both Gram-negative and Gram-positive bacteria [1]. For example, the *E. coli* hemolysin HlyA has a signal sequence at the *C*-terminus and is secreted by a type 1 secretion system (T1SS) in a single step directly from the cytosol to the extracellular space [40,41]. However, ClyA lacks a canonical signal sequence and undergoes no *N*-terminal processing during secretion [24,39]. In addition, substantial amounts of periplasmic proteins have been detected in culture supernatants from ClyA-overproducing cells [39], implying that ClyA secretion is accompanied by leakage of periplasmic contents. These findings indicated that ClyA is not secreted by general secretion machinery, such as a T1SS system, raising the question of how ClyA reaches the extracellular space.

In addition to known secretion systems, most bacteria also utilize membrane vesicle–mediated transport to secrete various toxins and deliver them to target cells. Bacterial membrane vesicles are spherical blebs with an average diameter of 20–300 nm that are naturally released from both Gram-negative and Gram-positive bacteria, and are often referred to as outer membrane vesicles (OMVs), especially in Gram-negative bacteria [42,43,44]. Recent studies have identified many bacteria-derived biomolecules, including toxin proteins, as OMV cargoes, and it has been reported that OMVs and their cargoes induce various biological effects in host cells [45,46,47,48]. For example, heat-labile toxin, one of the main enterotoxins produced by enterotoxigenic *E. coli*, is secreted via OMVs [49]. OMVs released by enterohaemorrhagic *E. coli* O157 cause cell death by delivering a cocktail of virulence factors, such as Shiga toxin 2a, cytolethal distending toxin V, and flagellin, into host cells [50]. ClyA has been identified as being an *E. coli* OMV cargo protein that is exported in a manner independent of canonical secretion systems [20]. Wei et al. demonstrated that ClyA is exposed on the surface of bacterial cells and accumulates in OMVs using immunofluorescence, electron microscopy, and atomic force microscopy. They concluded that the ClyA protein is translocated to the periplasmic space in its monomeric (inactive) form, and then oligomerizes to form active pore assemblies within the OMVs (Figure 1). This OMV-ClyA has been recently applied as a powerful tool in biomedical and bioengineering fields, as described in further detail below.

### 2.4. Assembled Form

A common feature of PFTs is their ability to assemble from a soluble, monomeric state into oligomeric, annular membrane complexes [7,51]. ClyA is one of the very few α-PFTs for which high-resolution structural information is available for both the soluble monomer and the annular pore complex [14,16,52,53]. ClyA is therefore considered the prototype of a subfamily of α-PFTs (often referred to as the ClyA family), and its structure has been well-characterized in numerous reports.

Briefly, the ClyA monomer consists of two domains: a tail domain composed of a bundle of five α-helices, and a head domain with a small β-hairpin (β-tongue) flanked by two short α-helices [14,16]. Molecular analysis has indicated the importance of the *N*- and *C*-terminal regions, and a partial hydrophobic sequence, in the process of ClyA translocation to the periplasm. In addition, a long α-helix (αA1) is crucial for transmembrane channel formation and function [54]. As shown in Figure 1, once ClyA reaches the target membrane, the monomer first undergoes a conformational change to a protomer and then predominantly assembles into an octameric [53] or 13-meric (redox state–independent) [55] pore complex. Alternatively, soluble ClyA monomers can assemble into 400-kDa dodecameric pores after binding to the detergent, *n*-dodecylmaltoside [14]. The structural mechanism of ClyA and ClyA-like pore formation has been well described in previous reports and reviews [17,51,56,57,58].

## 3. Mode of Action of the ClyA Toxin as a Virulence Factor

ClyA is a widely known PFT. Unlike *E. coli* HlyA and EhxA, which are posttranslationally modified by the addition of fatty acids [59,60,61], ClyA does not require posttranslational modification to be active. However, ClyA activity is tightly controlled at the transcriptional level under certain conditions and by regulation systems involving several regulators, as described above (see Section 2.2), and is also influenced by structural changes or environmental signals.

As described in Section 2.3, ClyA oligomerization is essential for full activity. For example, the structural change from the monomeric to the oligomeric form is crucial for ClyA activity and is involved in the altered redox status [20]. However, another study has shown that the intrinsic hemolytic activity of ClyA is independent of its redox state, and that assembly of both reduced and oxidized ClyA into the ring-shaped oligomer is triggered by contact with membrane lipids or a detergent [55]. ClyA harbors a cholesterol recognition and consensus (CRAC) motif, which typically stabilizes structural intermediates in assembly pathways that occur in the presence of cholesterol. The ClyA region containing this CRAC motif, which comprises residues L24-Y27-K29 (*N*-terminal helix), is conserved in ClyA homologs from several bacterial species (*E. coli*, *S.* Typhi, *S.* Paratyphi, *Bacillus subtilis*, and *S. sonnei*), and the presence of cholesterol stimulates pore formation by selectively stabilizing a protomer-like conformation, leading to hemolysis [62].

ClyA-mediated cell lysis involves a complex series of steps in which ClyA must recognize and bind to the target cell and then assemble to form a functional pore. A complete prepore is assembled on the target membrane (or, in the non-classical pathway, a soluble prepore forms within the OMVs); the membrane then becomes distorted, and a functional pore is formed by the insertion of α-helices into the lipid bilayer, resulting in hemolysis-like cell lysis [14,16,20,52,53]. In addition to hemolysis, it has been reported that ClyA has other cytotoxic effects on target cells. Oscarsson et al. demonstrated experimentally that highly purified ClyA exhibited cytotoxic activity toward J774 murine macrophage-like cells, which was associated with morphology changes and a substantial decrease in the detachment of target cells [63]. They speculated that membrane cholesterol may stimulate ClyA lytic action, and this effect was investigated in a subsequent study conducted by Sathyanarayana et al. [62]. Another group also demonstrated that purified ClyA and a ClyA-expressing *E. coli* strain were cytotoxic to both human and murine macrophages, and that this cytotoxicity was dose and time dependent. Their findings also suggested that ClyA induces massive apoptosis, as indicated by host-cell DNA fragmentation [64]. Thus, the pore-forming activity of ClyA is responsible for the induction of apoptosis in target cells, similar to the pore-forming *S. aureus* α-toxin and *E. coli* HlyA [65,66]. In addition, Fuentes and coworkers showed that ClyA assisted *S.* Typhi to invade human epithelial cells in vitro and promoted the colonization of deep organs in mice when expressed heterologously in *S.* Typhimurium [67].

Toxins can have different biological effects on target cells, depending on the concentration. Sublytic concentrations of some bacterial hemolysins can induce variations in the concentrations of eukaryotic second messengers that result in changes in host cell physiology, evidenced by changes in cellular morphology, as well as other responses [68,69,70,71]. At high concentrations, ClyA induces cytolysis of nucleated target cells, which is useful for lysing cells (e.g., epithelial cells) to allow bacterial cells to penetrate tissues at the intestinal phase of infection. However, at low (non-lytic) concentrations, ClyA can affect intracellular signaling processes regulating physiological responses. The low concentrations of ClyA delivered by OMVs affect Ca^2+^ homeostasis in epithelial cells by inducing slow intracellular Ca^2+^ oscillations, a signaling event that is linked to the expression of proinflammatory cytokines [69]. The ClyA protein and its pore assemblies are non-immunogenic, while proinflammatory responses can be induced by LPS (TLR4-dependent) or other components present in the OMVs. However, a recent report showed that ClyA exerts an indirect effect on innate immune signaling pathways by promoting the secretion of LPS-induced IL-1β in human macrophages through the TLR4 signaling and NLRP3-inflammasome pathways [72]. At present, our understanding of the biphasic activity of this toxin and its related physiological effects, which are the result of multiple factors, including extra-environmental signals, structural changes, and transcriptional regulation, is incomplete. The detailed mechanism by which ClyA contributes to bacterial virulence in vivo and in vitro needs to be addressed in further studies to elucidate the physiological role that this toxin plays in the process of bacterial infection.

## 4. Potential Applications in Nanopore Technology, Vaccine Development, and Tumor Therapy

Bacterial toxins clearly contribute to pathogenicity as virulence factors, but also have the potential for use in biotechnological applications [2,73]. ClyA is a virulence factor in terms of bacteriology, as described above. However, ClyA has also been used in various research fields, including biotechnology, and the number of studies related to this aspect has recently increased. In this section, recent research into the use of ClyA in biomedical and bioengineering applications, and future directions in these fields, are discussed.

### 4.1. Nanopore Technology

The structural properties (protein channels) of PFTs, including ClyA, are widely utilized as biological “nanopore” systems that enable single-molecule analysis without labeling, chemical modification, or surface immobilization [73,74]. In 1996, Kasianowicz et al. established the proof of concept of nanopores, an architectural type of biological machinery, using a membrane-embedded α-hemolysis channel [75]. One of the most recognized examples of the use of nanopores is the “MinION” sequencer (Oxford Nanopore Technology Ltd., Oxford, UK), which is integrated with a nanopore, and is used for genome DNA sequencing to obtain long read sequences [76,77,78]. Although the quality of the obtained sequences at present is not high compared with that of short reads from the Illumina platform, this nanopore sequencing technology enables simpler and more rapid determination of complete genome sequences, especially for organisms with small replicon sizes, such as bacteria and viruses [79,80,81,82].

The Maglia group first demonstrated that ClyA can accommodate natively folded proteins within its nanopore lumen [83] and have subsequently reported the utility of ClyA as biological nanopore for the detection of single macromolecules in multiple advanced studies [84,85,86,87,88]. In a study by Wang et al., ClyA was used to develop an electrode-free nanopore sensing method that enables optical monitoring of single-molecule events without any electrical connections [89]. Thus, recent studies have shown the potential for ClyA to serve as a biological nanopore, which in the future could be integrated into miniaturized, low-cost devices for use in medicine, industry, environmental monitoring, or for single-cell analysis.

### 4.2. Vaccine Development

ClyA is useful in vaccine development because surface-exposed or secreted antigens are often more immunogenic than cytoplasmic antigens, which is an important factor in developing effective vaccines to protect against pathogens. Galen et al. successfully engineered a recombinant protein comprising ClyA fused to the *Bacillus anthracis* protective antigen (domain 4 moiety) and demonstrated that this fusion protein was exported and exhibited immunostimulatory activity in mice [90]. Their findings showed that ClyA could be an attractive export system for displaying foreign (heterologous) antigens in attenuated bacterial strains, which paved the way for exploring this feature of ClyA in numerous subsequent studies.

Importantly, ClyA is secreted by OMVs, and is therefore a good fusion partner and an ideal scaffold for delivering exogenous proteins or antigens to OMVs, which are then presented on the surface of released OMVs. Kim et al. demonstrated that fully functional, recombinant beta lactamase, GFP, and anti-digoxin antibody fused to the *C*-terminus of ClyA were secreted in OMVs. This work highlighted the flexibility of recombinant protein display on the OMV surface [91]. A subsequent study showed that an antibody response directed against GFP was induced in mice immunized with OMVs expressing recombinant GFP fused to ClyA [92]. These seminal studies opened a new research avenue for the development of next-generation therapies that take advantage of the genetic flexibility offered by the ClyA-OMV system.

The engineered ClyA-OMV system has been recently applied in the development of vaccines against various pathogens, including SARS-CoV-2 [93]. For example, it has been reported that the successful expression and localization of ClyA-M2e fusion proteins in OMVs led to protection against a lethal dose of H1N1 influenza A virus [94]. Another successful example of the use of ClyA in vaccine development was a study in which a complete *Acinetobacter baumannii* Omp22 protein fused to ClyA was displayed on the surface of *E. coli* OMVs. Immunization with the engineered OMVs induced a strong antigen-specific humoral immunity response in mice and protected mice from a lethal challenge with a clinical *A. baumannii* isolate [95]. During the ongoing COVID-19 pandemic, messenger RNA has been widely used to generate immunogens in SARS-CoV-2 vaccines, which have been very effective in preventing COVID-19 [96,97,98,99]. However, the current vaccines against SARS-CoV-2 have several issues, including low yields, transportation difficulties, high manufacturing costs, and complex and delicate manufacturing processes. In recent work, Yang et al. reported the development of bacterial biomimetic vesicles (BBVs) loaded with a fusion protein comprising ClyA and the SARS-CoV-2 spike protein receptor binding domain (RBD), called RBD-BBV [100]. Combining this strategy with high-pressure homogenization technology further enhanced the presentation of the RBD on BBVs and the yields of RBD-BBVs. When injected subcutaneously, these RBD-BBVs accumulated in lymph nodes, promoted antigen uptake and processing, and elicited SARS-CoV-2-specific humoral and cellular immune responses in mice. Thus, it is expected that the ClyA-OMV system could be used in the development of vaccines against emerging pathogens, such as SARS-CoV-2.

### 4.3. Tumor Therapy

Historically, bacterial infection and bacterial products, including toxins, have been used as challenges in cancer treatment. For example, Coley’s toxin, which is a cocktail of heat-killed bacteria, has been used to stimulate an anti-cancer immune response [101,102]. The ClyA protein alone and the ClyA-OMV system have been investigated as potential anti-tumor agents or tools. Ryan et al. first demonstrated that the cytotoxic properties of ClyA, in combination with a hypoxia-inducible promoter, increased necrosis in hypoxic regions of tumors and inhibited tumor growth in mice, indicating the potential of ClyA as an anti-tumor agent [103]. It has been suggested that *E. coli* OMVs have a potential role in tumor immunotherapy, as intravenous injection of these OMVs induced strong IFN-γ- and T cell-mediated anti-tumor responses [104]. In another study, the use of a ClyA-OMV system successfully achieved presentation of the ectodomain of the programmed cell death protein 1 (PD-1). The engineered OMVs bound to the PD-1 ligand 1 on tumor cell surfaces, thereby facilitating PD-1 ligand 1 internalization and reducing the expression levels on tumor cells [105]. A very recent study by Thomas et al. showed that ClyA served as an anchoring protein for hyaluronidase (Hy) in hypervesiculating *E. coli* Nissle-produced OMVs and potentially exerted a cytotoxic effect against cancer cells [106]. These engineered OMVs helped distribute Hy more effectively and uniformly inside the tumors, permitting the penetration of drugs and potentiating their effects by considerably decreasing the levels of hyaluronic acid in the tumor tissues. These results indicated that the ClyA-OMV system could be an effective tool for stromal remodeling, cytolytic therapy, and improving the activity of anti-cancer targeted therapeutics. Similar to this approach, ClyA has been used as a leader protein for presenting human epidermal receptor 2 (HER2; expressed at high levels in breast cancer cells) affibodies on the surface of OMVs. The engineered OMVs delivered a loaded therapeutic siRNA cargo to ovarian and breast cancer cells expressing HER2, subsequently inducing cancer cell death without appreciable side effects [107]. In a recent study using advanced techniques, a flexible tumor vaccine platform based on OMVs expressing ClyA as a fusion partner for tumor antigens in conjunction with a “plug-and-display” system [108] comprising tag/catcher protein pairs [109] was established. This approach enabled the display of various tumor antigens linked to protein tags on the OMV surface, rapidly and simultaneously, as the protein catchers were fused to ClyA. The OMV-mediated tumor antigens were effectively delivered to lymph nodes and then presented to dendritic cells, resulting in the induction of antigen-specific T-lymphocyte-mediated anti-tumor immune responses in murine tumor models.

Cancer is a global health issue, and researchers continue to explore strategies for cancer treatment from various perspectives. The ClyA-OMV system could further accelerate research in this field and help develop a new generation of more effective cancer therapies.

ClyA can be fused to a foreign protein and stably anchor it on the surface of bacterial cells. This feature enables the facile functional display of a variety of different prokaryotic and eukaryotic proteins. In addition, ClyA elicits this ability to the fullest in combination with OMVs. For example, surface-exposed antigens facilitated using ClyA are more effectively immunogenic than cytoplasmic antigens and thus are useful for vaccine development [90,95,100]. In tumor therapy, ClyA serves as a fusion partner for tumor antigens and can form part of the OMV-based delivery platform for anti-tumor agents against specific cancer cells [105,106,107]. Although the ClyA protein has the potential for versatile applications, the biphasic activity of this toxin and its related physiological effects in vivo remain unclear. In addition, OMVs enclose various bacterial proteins and other molecules (e.g., LPS and outer membrane proteins) that might have unwanted biological effects on normal cells [46,47]. Therefore, the potential side effects, including the intrinsic toxicity of ClyA, must be carefully considered in the future development of platforms using ClyA, especially in vaccine development and tumor therapy.

## 5. Conclusions

The properties of ClyA, including its function, activity, secretion, and structure, have become increasingly well characterized since its discovery a quarter of a century ago. However, more work is needed to investigate the physiological roles of ClyA and understand the detailed effects of this toxin on host cells. Determining the structure of ClyA and how it is secreted were noteworthy findings that had a substantial impact on subsequent studies. While bacterial toxins are major virulence determinants that can cause considerable damage to the host, understanding how they function can also lead to the development of various practical applications that take advantage of the toxin’s unique characteristics. Recent studies have demonstrated the ways in which ClyA’s unusual features can be exploited to develop cancer therapies, nanopores, and vaccines, showing its versatility. Other known or newly identified toxins may also be as useful as ClyA in applied fields, and I hope that this review will inspire future studies that expand the utility of bacterial toxins.

## Figures and Tables

**Figure 1 toxins-14-00078-f001:**
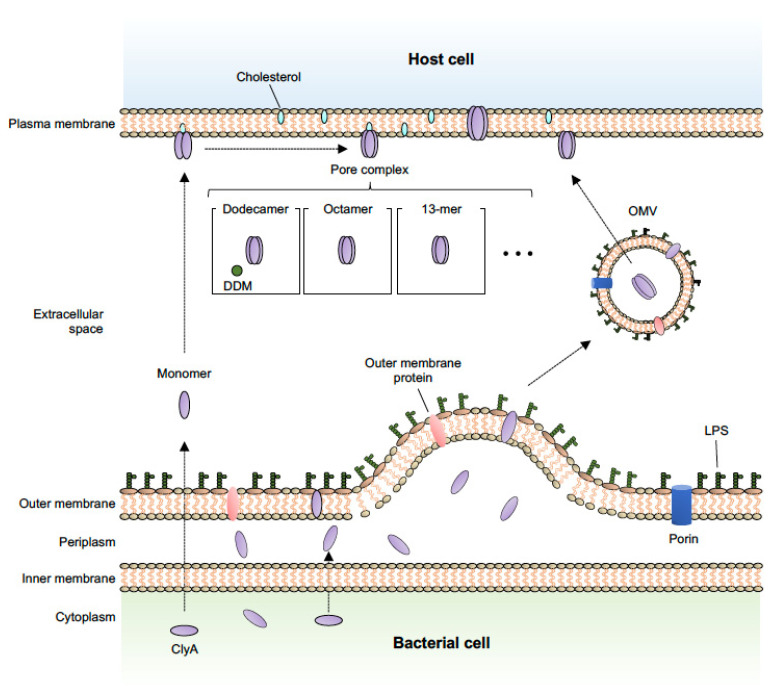
Schematic representation of the OMV-mediated secretion pathway for ClyA and the predicted pore assembly pathway. ClyA protein is secreted by OMVs and the monomer oligomerizes to form the active pore assemblies within OMVs in a redox state-dependent manner [20]. In the assembly process, once ClyA reaches the target membrane, the monomer first undergoes a conformational change to a protomer and then predominantly assembles into an octameric [53], 13-meric (redox state-independent) [55], or dodecameric [14] pore complex by contact with membrane lipids or the detergent, *n*-dodecylmaltoside (DDM).

**Table 1 toxins-14-00078-t001:** List of major transcriptional regulators for *clyA* gene expression.

Gene	Product/Function	Organisms	Relevant Descriptions	References
*hns*	heat-stable nucleoid-structuring protein	*E. coli*	strongly repress the transcriptional expression of *clyA* gene under laboratory condition	[12,13,28,29,39]
*slyA*	MarR-family transcriptional regulator	*E. coli* *S.* Typhimurium	activate the expression of *clyA* gene by antagonizing the H-NS–mediated transcriptional repression when overexpressing SlyA	[12,13,28,29,39]
*mprA*	MarR-family transcriptional regulator	*E. coli*	activate the expression of *clyA* gene by antagonizing the H-NS–mediated transcriptional repression when overexpressing MprA	[24]
*fnr*	fumarate and nitrate reduction regulator	*E. coli*	activate the expression of *clyA* gene in response to oxygen depletion	[33,35]
*crp*	cyclic AMP receptor protein	*E. coli*	activate the expression of *clyA* gene in response to glucose starvation	[33]
		*S.* Typhi	repress the transcriptional expression of *clyA* gene (on SPI-18) via down-regulating *rpoS*	[37]
*phoP*	transcriptional regulator (two-component regulatory system PhoP/PhoQ)	*S.* Typhi	up-regulate the expression of *clyA* gene (on SPI-18) via *rpoS* under low pH and low Mg^2+^	[36,38]
*fis*	DNA-binding protein	*S.* Typhi	down-regulate the expression of *clyA* gene (on SPI-18) in CRP-independent glucose-dependent manner	[38]
*rpoS*	RNA polymerase sigma factor	*S.* Typhi	relate to the transcriptional upregulation of *clyA* gene (on SPI-18) in low pH and high osmolarity condition (predicted central regulator in the *clyA* regulatory network)	[36,38]

## Data Availability

Data sharing not applicable.

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
