# Peer review of "Cytolysin A (ClyA): A Bacterial Virulence Factor with Potential Applications in Nanopore Technology, Vaccine Development, and Tumor Therapy"

_toxins, 2022, doi:10.3390/toxins14020078_

Round 1

Reviewer 1 Report

The manuscript “ Cytolysin A (ClyA): a bacterial virulence factor with potential utility for biotechnology applications” is an interesting review which provides a useful characterization of the ClyA and its possible future application in biotechnologic-clinical fields.

Although the review has a certain merit and the points discussed are topical, it is possible to move major and minor criticisms:

  • Major points.
  1. The abstract has to take into account more aspects of the intoduction, paragraph 2 and 3.
  2. The paper completely lacks tables and figures: Authors should add a table to resum main infos for “transcriptional regulation” and cartoons for “the secretion pathway” and assempled form”, for a clearer and more captivating reading.
  3. Since the biotechnologic part is strictly linked to clinical possible, future applications, please insert also a paragraph (or add info in a preexisting paragraph) discussing more extensively this part (advantages and disadvantages of bacterial toxins in cancer therapy)

  • Minor points
  1. Pag 1, line 31: please adjust the bibliographic insertion.
  2. Please insert this paper: (Toxins201810(6),239; https://doi.org/10.3390/toxins10060239) among the citations (42-43); line 127, pag3.
  3. The authors should rewrite the sentence at pag 6, line 290-291
  4. The review could be enriched by adding this citation: J. Mol. Sci. 2020, 21(6), 2207; https://doi.org/10.3390/ijms21062207 , among the citations (67-69), pag.5, line 203
  5. The last paragraph “Nanopore technology” should be moved to the first position (4.1)

Author Response

#Reviewer1

The manuscript “ Cytolysin A (ClyA): a bacterial virulence factor with potential utility for biotechnology applications” is an interesting review which provides a useful characterization of the ClyA and its possible future application in biotechnologic-clinical fields.

Although the review has a certain merit and the points discussed are topical, it is possible to move major and minor criticisms:

Thank you for the valuable comments and suggestions on the manuscript. Please see below, in blue, for a point-by-point response to the reviewers’ comments and concerns.

  • Major points.
  1. The abstract has to take into account more aspects of the intoduction, paragraph 2 and 3.

I have made revisions of the abstract accordingly.

  1. The paper completely lacks tables and figures: Authors should add a table to resum main infos for “transcriptional regulation” and cartoons for “the secretion pathway” and assempled form”, for a clearer and more captivating reading.

Thank you. I have added one table (Table 1. A list of major transcriptional regulators for clyA gene expression) and one figure (Figure 1. Schematic representation of the OMVs-mediated secretion pathway for ClyA and the predicted pore assembly pathway) in the revised paper (Page 3, Line 113-114; Page 5, Line 168-175).

  1. Since the biotechnologic part is strictly linked to clinical possible, future applications, please insert also a paragraph (or add info in a preexisting paragraph) discussing more extensively this part (advantages and disadvantages of bacterial toxins in cancer therapy)

Thank you. I totally agree with you. The relevant statements have been added to the end of “Section 4” (Page 8, Line 348-361).

  • Minor points
  1. Pag 1, line 31: please adjust the bibliographic insertion.

Thank you. I have made revisions accordingly.

  1. Please insert this paper: (Toxins201810(6),239; https://doi.org/10.3390/toxins10060239) among the citations (42-43); line 127, pag3.

The reference has been cited in the text of revised manuscript.

  1. The authors should rewrite the sentence at pag 6, line 290-291

I have re-written the sentence in the revised manuscript (Page 7, Line 324-326).

  1. The review could be enriched by adding this citation: J. Mol. Sci. 202021(6), 2207; https://doi.org/10.3390/ijms21062207 , among the citations (67-69), pag.5, line 203

The reference has been cited in the text of revised manuscript.

  1. The last paragraph “Nanopore technology” should be moved to the first position (4.1)

This subsection has been rearranged accordingly.

Reviewer 2 Report

Review paper entitled “Cytolysin A (ClyA): a bacterial virulence factor with potential utility for biotechnology applications” focused on the brief introduction of ClyA, the encoding gene and its transcriptional regulation, the structure assembly. Based on the basic information about ClyA, the application in the fields of vaccine development, tumor therapy and nanopore-related are the key content in this review work.

The minor and major points are listed below for references.

Minor

Title: Cytolysin A (ClyA): A bacterial virulence factor with potential utility for biotechnology applications.  After the colon, the first letter “A” should be capitalized.

Line 51-55. Will do (simple future tense) is not proper. Besides, pay attention to the tense in the whole manuscript.

Line 72 passive or active sentence?

Line137, …concluded that……which is……

Line 150-151, …revealed that…are…

Line 161 “ClyA is a widely known as PFTs”, delete a

Line 218-220 this sentence needs to be re-written.

Line 221 the title “4. Potential applications in biotechnology”. And the title of this review need to be given with more proper words.

Line 226-234 these three unusual features need to be re-arranged.

Major

  1. About the part 4, this is the key of your review, I think. So besides it can be applicated in vaccine development, tumor therapy and the development of nanopore technology, it is better to state some advantages and disadvantages with a few sentences.

  1. need proof-reading of English-native speakers

Reviewer 3 Report

Dear Authors,

Below I present my comments/report on the submitted manuscript entitled "Cytolysin A (ClyA): a bacterial virulence factor with potential utility for biotechnology applications".

In general, the work seems to me to be properly conducted, and the text of the manuscript refers to the proposed title. However, it is necessary to make some corrections: minor in terms of content and major in terms of editorial.

  1. in abstract (lines 4-5) – should be clarified, i.e. [(…) by some bacteria from the Enterobacteriaceae family]. The same in line 34.
  2. in lines 5, 34 and 76 should be [Enterobacteriaceae] instead of [Enterobacteriacea]
  3. in line 31 – [(Parker et al., 1989)] is not necessary
  4. in lines 36, 87 and 177 – please write the names of Salmonella serovars with a capital letter and without italics (e.g. Salmonella Typhimurium). In addition, on line 177 in the name S. Paratyphi there should be a dot after the name Salmonella (now there is a comma).
  5. in line 37 – please add [E. coli strain K-12]
  6. in line 63 should be [clyA gene] instead of [clyA genes]
  7. please always write gene names in italics, e.g. clyA in lines 71, 72, 99, 103, 104, and 107
  8. In the case of citing the authors, the year of publication in brackets (or not) should be standardized. If the Authors want to enter the year of publication, this should be completed in lines 70, 134, 169, 174, 191, 197, 238, 247, 257, 267, 281, 286, 290, 297, 302, 327, and 330.
  9. The name of Gram-negative rods Salmonella Typhimurium is a shortened name and of course we know that it is Salmonella enterica subsp. enterica serovar Typhimurium. The abbreviated name of these rods was written on line 36 (first time, then on lines 70-71). After, on lines 72-73 the full taxonomic name of Salmonella was given (incorrectly spelled: it should be Salmonella enterica subsp. enterica serovar XXX). I suggest you write down the full name of serovar at the first possible moment, and then the short names.
  10. in lines 70-71 – S. Typhi and S. Paratyphi A are serovars, not strains.
  11. in line 73 – should be [in various non-typhoidal (…)] instead of [in various non-typhoid (…)]
  12. in line 74 - the name of the subspecies should be in italics, e.i. S. enterica subsp. arizonae
  13. in line 75 - I suggest adding [gene] between [clyA] and [and its (…)]
  14. in line 100 – please write [Salmonella] in italics
  15. in line 126 – the Authors do not need to explain the OMVs again - this is explained in the Abstract
  16. in line 177 – should be [Bacillus subtilis] instead of [Bacillus subtilus]
  17. in line 191 – [in 2018] is not necessary, the year of this publication will be entered on line 174
  18. in line 191 - I cannot find any works by Uhlin et al. in the references. Is it a missing reference item, or was it publication No.69 (Uhlen et al.)?
  19. in line 209 – should be [Ca2+] instead of [Ca2+]
  20. in line 214 - maybe it will be better [IL-1β] instead of [IL-1beta]
  21. in lines 219-220 – please write [in vivo or in vitro] in italics
  22. In some parts of the manuscript there are no references, e.g. in lines 216-234.
  23. in line 231 - please remove the double space between [(…) for display,] and [3) the ClyA (…)]
  24. Why in line 238 the Authors cited Green et al. as the first author instead of Galen et al. (2004); the same in line 247 is Putnam et al. instead of Kim et al. (2008), in line 282 is Green et al. instead of Ryan et al. (2009), in line 327 is Maglia et al. instead of Soskine et al. (2012)?
  25. in line 262 – please write [A. baumannii] in italics
  26. References are completely inconsistent with the journal's requirements. In addition, various references to the literature have been used, in the titles of scientific reports the names of genes, the names of bacteria are written without italics.

The article also lacks the presentation of the content in the form of a table or figure. If possible, adding a graphical element will enrich this interesting article.

Round 2

Reviewer 1 Report

The authors have addressed the points moved by the reviewer.